

# Farmers' perception of the ecosystem services provided by diurnal raptors in arid Rajasthan

Govind Tiwari[1], Puneet Pandey[2,3], Rahul Kaul[4] and Randeep Singh[1]

[1] Amity Institute of Forestry and Wildlife, Amity University, Noida, Uttar Pradesh, India
[2] Enprotec India Foundation, Dehardun, Uttarakhand, India
[3] Conservation Genome Resource Bank for Korean Wildlife (CGRB), Research Institute for Veterinary Science and College of Veterinary Medicine, Seoul National University, Seoul, Republic of Korea, Seoul, Seoul, Republic of Korea
[4] Wildlife Trust of India, Noida, Uttar Pradesh, India

## ABSTRACT

Farmers are the most important stakeholders in wildlife conservation in the agricultural landscape. Understanding the farmer's perceptions, attitude, behaviour, and knowledge toward conservation is critical in developing an effective conservation programme in human-dominated landscapes. We conducted semi-structured face-to-face interviews with 373 farmers to understand the farmer's perception of ecosystem services provided by diurnal raptors in the arid region of Rajasthan from July 2020 to February 2021 and from August 2021 to January 2022. We grouped ecosystem services and disservices into larger categories and estimated the correlation between them, finding that disservices are negatively correlated with benefits. Raptors were perceived as beneficial for their role in controlling rodents and pests, but negatively for poultry predation. In addition, we built a binomial generalised linear model with a logit function to better understand the factors that influence farmers' perceptions of raptors (positive or negative). We observed that males and females have different attitudes toward the ecosystem services provided by raptors. It is critical to understand social perceptions in order to conserve species that are rare on a global scale but may face negative perceptions on a local scale. Our study connects ecological information with socio-demographic factors, which can be useful in developing policy measures for raptor conservation.

## INTRODUCTION

The importance of the social-ecological perspective or social dimensions (*i.e.*, attitudes, beliefs, perceptions or values) in human-dominated agricultural landscapes are now shaping biodiversity conservation and informing policymakers and land-use managers around the world to take appropriate decisions (*Bennett et al., 2015*; *Pooley et al., 2017*; *Morales-Reyes et al., 2018*). Agriculture constitutes the dominant land use in many countries (*Cai & Pettenella, 2013*) with agricultural landscapes providing refuge and habitat for a variety of wildlife species (*Perrings et al., 2009*) like chital deer, chinkara,

Corresponding author
Randeep Singh,
randeep04@rediffmail.com

granivorous and omnivorous birds and butterflies (*Karanth & Kudalkar, 2017*). The loss of biodiversity in agricultural landscapes is linked to the loss of benefits obtained from ecosystems (*Perrings et al., 2009*; *Morandin & Winston, 2006*). As a result, policymakers often encourage farmers to protect the habitat of many threatened species on their farms (*Kross et al., 2018*). These relations make it critical for researchers to comprehend the relationship between benefits obtained from ecosystem services and biodiversity conservation in agricultural areas (*Gorosábel, Bernad & Pedrana, 2022*).

Many species (*i.e.*, insects, birds, and rodents) are known crop pests in agricultural landscapes; they directly cause harm to farmers by damaging crops (*Johnson et al., 2018*), which may result in reduced productivity or increased production costs (*Zhang et al., 2007*; *Şekercioğlu, Wenny & Whelan, 2016*; *Garcia et al., 2010*). Raptors or birds of prey, on the other hand, are highly valued in some agroecosystems because they can significantly control pest abundance or activity and act as intraguild predators. They offer biological crop pest control and particularly those of insects (*Belaire et al., 2015*; *Kross et al., 2018*; *Shave et al., 2018*; *Garcia et al., 2010*), which benefits farmers indirectly (*Kross et al., 2016*) as raptors control insects and rodents and reduces the input cost of farmers. Raptors can play an important role in reducing pesticide usage in farms which will not only provide economic benefits but also reduce pest outbreaks (*Naranjo, Ellsworth & Frisvold, 2015*). Raptors also have a positive socio-ecological impact by lowering human health risks and preserving biodiversity (*Gibbs, Mackey & Currie, 2009*; *Sarwar, 2019*).

On the other hand, raptors face significant threats in agricultural landscapes due to a variety of anthropogenic activities like intensive agriculture practices, use of pesticides to maintain food production, land use change, deforestation, habitat alterations, hunting, poisoning by anticoagulant rodenticides (*Hindmarch, Rattner & Elliott, 2019*) and trade (*Gibbs, Mackey & Currie, 2009*). Raptors are frequently persecuted for their negative effects on property, livestock and human life (*O'Bryan et al., 2022*). Indeed, anthropogenic threats are cited as one of the major causes of decline in the ecological or ecosystem services provided by raptors around the world (*Emmerson et al., 2016*; *Rusch et al., 2016*). Raptor conservation in agricultural landscapes therefore depends to a great extent on farmers' knowledge, behaviour, farm practices, and attitudes (*Horgan, Mundaca & Crisol-Martínez, 2021*; *Nyirenda et al., 2017*). As primary stakeholders, farmers' direct and indirect involvement in the decision making for raptor conservation is critical. It is also prudent to note that farmers can provide important information about raptor distributions, breeding, threats, and ecosystem services in agricultural landscapes (*Gaston et al., 2018*; *Kross et al., 2018*) and it can thus be inferred that perception of farmers to a great extent can determine the species' conservation outcomes.

Cultural ecosystem services, includes the environmental basis for aesthetic, spiritual, and recreational experiences, cultural heritage, sense of place and ways of life (*Winthrop, 2014*). According to *Zoeller, Gurney & Cumming (2022)*, there is a well-established understanding of the relationships between functional traits of organisms and provisioning and regulating ecosystem services. However, the specific traits that contribute to the benefits derived from cultural ecosystem services are not yet clearly identified and the contribution of raptors to cultural ecosystem services such as sense of place or education is

unknown (*Echeverri et al., 2019*). *Horgan, Mundaca & Crisol-Martínez (2021)* explain that the concept of sense of place encompasses people's emotional attachment to a particular location, which is frequently visited by tourists and researchers for observing a specific species. It encompasses various aspects of individuals' perceptions and interpretations of the environment, including their attachment, identity, and the symbolic meaning they attribute to the place. Sense of place has the potential to bridge social and ecological issues, recognizing the interconnectedness between human experiences and the ecological context of a specific area. Also, education and training programs to train nature guides and people operating home stays for tourists were organized by forest department and non-government organizations. We believe respondents in the study region might relate to such cultural ecosystem services, especially those who live near protected areas or community protected grasslands.

The current study aims to understand farmers' perceptions of raptors in an arid region of the state of Rajasthan, India, as well as the socioecological factors that influence these perceptions about the raptors. Organic agriculture has gained popularity in the arid region of Rajasthan in recent years (*Dangour et al., 2010*) and its potential in rural communities is recognised (*Panwar et al., 2010*). Farmers who grow organic crops have fewer pests control options (*Costa et al., 2019*; *van Bruggen, Gamliel & Finckh, 2016*) and raptors can serve as a natural biological pest control agents in such situations (*Shave et al., 2018*; *Garcia et al., 2010*). Therefore, considering the afore-mentioned points, we hypothesised that raptors are perceived to be more beneficial to farmers growing organic crops in the region.

Furthermore, rural communities in the study area rely primarily on agriculture for a living, with small poultry operations supplementing household income (*Ithika, Singh & Gautam, 2013*). Male farmers are primarily responsible for livelihood and outdoor work (tourism, agriculture, crop protection, and animal husbandry/livestock grazing) whereas female farmers are responsible for the household work, livestock management at the sheds in homes, fodder/wood collection, and poultry management (*Kumar et al., 2021*). We therefore hypothesized that male farmers interacted with raptors more often and distinctly than female farmers and thus have different perceptions about raptors.

In addition, we also assessed farmers' attitudes toward other species (bats and perching birds) using the same criteria. This study aimed to collect baseline data on the perceptions of farming community towards raptors for the benefit of forest department and policymakers for management of raptors in the agriculture ecosystems. Through community outreach programmes, the forest department and conservation organisations can initiate education awareness programmes to improve farmers' knowledge of ecology and ecosystem services provided by raptors in agriculture ecosystems for future conservation initiatives in the region.

## MATERIAL AND METHODS

### Ethical statement

To conduct the farmer perception survey, we have obtained the required approval from human ethics committee in our institute (Amity Institute of Forestry and Wildlife, Noida, Uttar Pradesh) letter no-AUUP/AIFW/REC/2020/01.

## Study area

In the hot arid region of Rajasthan, India, we have studied (*Tiwari et al., 2021*; *Tiwari et al., 2023*) community of diurnal raptors (Fig. 1). The study area covered 0.198 million square kilometres and is located between 24°31′ and 30°12′ north latitudes and 69°15′ to 76°42′ east longitudes. The region is characterised by low and erratic rainfall, with an average annual rainfall of 500 mm, ninety percent of which falls during the monsoon season (*Moharana et al., 2012*). Temperatures can range from 0 °C in the winter to 50 °C in the summer. The terrain is slightly undulating within the venue of sand deposited by inland drainage and streams with salt lakes and limited water resources and arable lands (*Sharma & Sharma, 2004*). Northern tropical thorn forests (Champion and Seth Classification 6B) which include *Calligonum polygonoidis*, *Prosopis cineraria*, *Prosopis juliflora*, *Acacia capparis*, *Acacia Senegal*, *Acacia catechu*, *Anogeissus pendula*, *Butea monosperma*, and *Azadirachta indica*, cover the rolling arid landscape. Anthropogenic activities have an impact on the landscape because 22.5 million people live here, making it the world's most populous desert at a density of about 84 people per square kilometre (*Singh & Kumar, 2015*). Most residents' occupations (60%) are farming, raising livestock and related activities (*Government of Rajasthan, 2023*). This area is home to numerous resident and migratory raptors despite its harsh climate and man-made limitations. The existence of so many raptor species in the arid region can be attributed to both socioeconomic and climatic factors (*Chhangani, 2007*).

# METHODS

## Data collection

From July 2020 to January 2022, we conducted in-person interviews with 373 respondents (Supporting Information-S1) using semi-structured questionnaires (File S1). People were informed of the purpose of the study before they participated in interviews and only then did, they give their informed consent (verbal). No two respondents were interviewed on the same farm. Response was recorded only *via* interviews and no other mode (video/audio) were used to obtain the response of the respondents.

There were three main sections to the questionnaire: (A) sociodemographic profile of the respondents, (B) details on how farmers feel about raptors and the ecosystem services they provide, and (C) details on how they feel about other species in the area. On a five-point Likert scale (1-Strongly disagree to 5-Strongly agree) (*Likert, 1932*), respondents were asked to rank raptors according to their perception about nine different statements. The statements were about the possible ecosystem benefits and harm that raptors may provide (*Martínez-Sastre et al., 2020*; *Echeverri et al., 2019*; *Zoeller, Gurney & Cumming, 2022*). Information related to poultry predation was also obtained from the respondents who are involved in poultry raising and management ($n = 104$).

To measure the perception of male and female respondents for ecosystem services provided by other avian species data was organised on a Likert scale (*Likert, 1932*) and then compared with the ecosystem services offered by the raptors. Our methodology is based on the study done by *Kross et al. (2018)*. The questions largely pertained to the trends in the
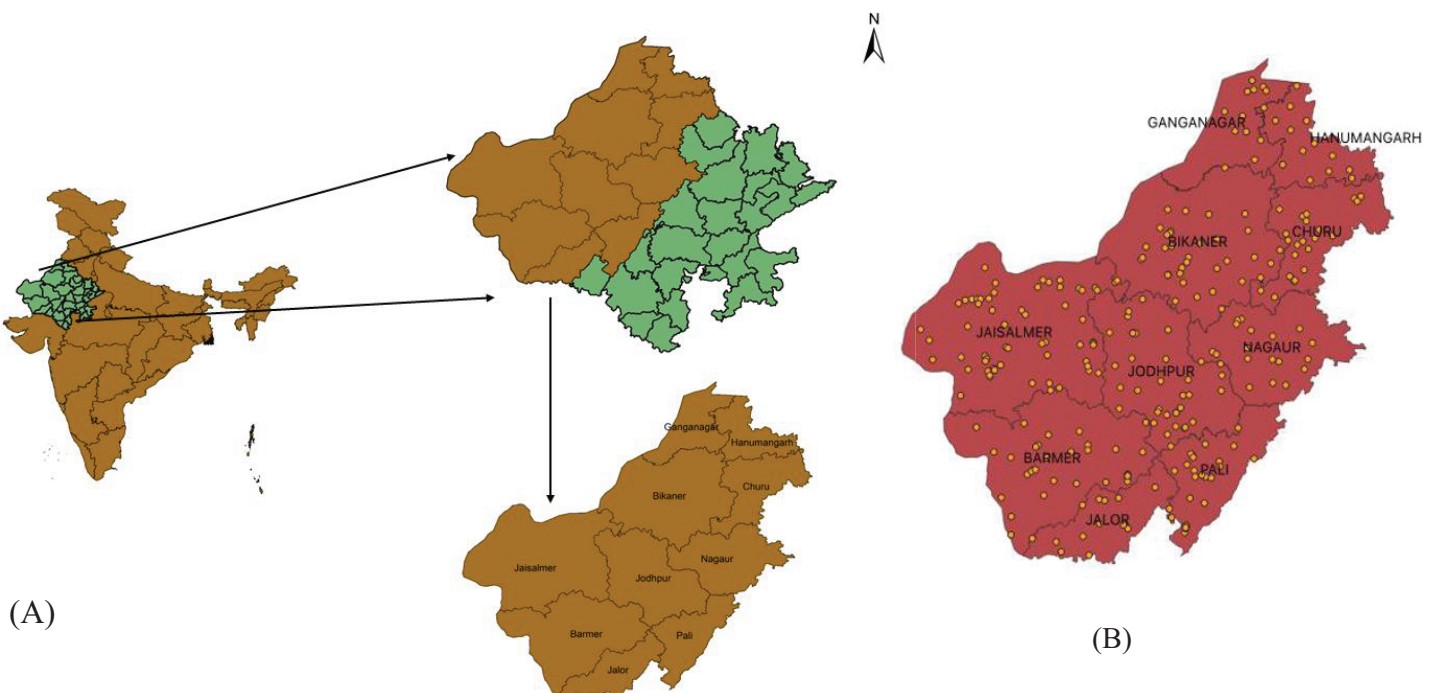

**Figure 1 Location map of (A) study area and (B) sampling location in arid region Rajasthan during July 2020 to February 2022.**

raptor population in the areas and were the trends increasing or decreasing and what is their overall perception of that species (*Morales-Reyes et al., 2018*). The survey was carried out utilising convenience sampling (*Som, 1995*). We have considered ($p = 0.05$) threshold to determine statistical significance in our analysis.

## Data analysis

We used regression analysis to examine the relationship between farmers' perceptions of species population trends and overall perception (Supporting Information-S2) of vultures and raptors (*Morales-Reyes et al., 2018*). The responses to various ecosystem services which were obtained on Likert scale were then divided into groups according to gender and broad categories (Table 1). Perception of respondents for the various services/disservices is also shown on a Likert scale. To verify the data's internal consistency, Cronbach's alpha for the variables was estimated (*Cronbach, 1951*). To reduce the dimensionality of the variables, explanatory factor analysis was performed on Likert scale which produced three different items that represented nine different services. Scree plots were used to estimate the number of factors (Supporting Information S3–S4).

Factanal function was used to divide the likert scale items into three major categories (Table 1) of ecosystem services provided by raptors (*Echeverri et al., 2019*; *Zoeller, Gurney & Cumming, 2022*). For both male and female respondents, we also calculated pairwise correlation across these nine ecosystem services. ANOVA with a *post-hoc*-Tukey Honest Significant Difference (HSD) test was employed to determine whether there was a statistically significant difference between the perceptions of the male and female

**Table 1 Categorization of perception of farmers on ecosystem services in major categories (*Echeverri et al., 2019*; *Zoeller, Gurney & Cumming, 2022*).**

| Serial code | Construct | Benefit and loss of raptors | Factor loading | | |
|---|---|---|---|---|---|
| A | Indirect benefits | A1. Increases crop quality | 0.787 | 0.150 | 0.252 |
| | (Cronbach's alpha = 0.913) | A2. Increases yield | 0.810 | 0.215 | 0.212 |
| | | A3. Essential for crop production | 0.904 | 0.209 | 0.192 |
| B | Negative/Disservice | B1. Causes damage to pollinators | 0.307 | | 0.587 |
| | (Cronbach's alpha = 0.769) | B2. Causes damage to poultry | 0.217 | 0.332 | 0.735 |
| | | B3. Causes damage to livestock | 0.127 | 0.353 | 0.662 |
| C | Direct benefits | C1. Controls insects | 0.137 | 0.769 | 0.132 |
| | (Cronbach's alpha = 0.813) | C2. Controls rodents | 0.197 | 0.734 | 0.237 |
| | | C3. Alternative to pesticides | 0.175 | 0.669 | 0.264 |

respondents about the ecosystem services that raptors provide. Tukey's HSD tests are conservative because they lessen the chance of a Type I error in addition to allowing comparisons between groups with multiple categories (*Abdi & Williams, 2010*; *Nanda et al., 2021*). Cohen's d (*Diener, 2010*) was used to estimate the effect size between male and female respondents.

We created a logit-based binomial generalised linear model (GLM) (*Luoto & Hjort, 2004*; *MacKenzie, 2018*). The sociodemographic data of respondent's was kept as an independent variable and their perception of raptors whether they were helpful or harmful, was kept as a dependent variable. To compare respondents' perceptions with those of the other species present in the area (bats and perching birds), perceptions of the respondents were also collected for those species (Supporting Information S5–S10). In the second GLM, data of farmers who owned chicken was examined. Perception of raptors (Beneficial or Harmful) was kept as a dependent variable and loss of chicken due to raptor predation and investment in chicken management along with socio-demographic information was kept as an independent variable (S11). The result of GLM is untransformed model output. The "CAR" (*Fox & Weisberg, 2019*), "ggplot2" (*Wickham, 2016*), "Psych" (*Revelle, 2022*), and "Corrplot" (*Wei & Simko, 2021*) packages were used to analyse all the data in R (*R Core Team, 2020*). The open-source, free QGIS software was used to prepare the location map (*QGIS.org, 2021*).

## RESULTS

### Perception of indirect benefits

There was a significant difference between male and female respondents' perceptions of how raptors affect crop quality and production ($p = 0.0007$ and $p = 0.0001$, respectively). Cohen's d value for both the categories were (0.361) and (0.540) respectively which suggests a subtle effect size difference. Regarding the impact of raptors on overall yield, there was no discernible difference between the opinions of the two categories of respondents ($p = 0.852$). Effect size was also merely statistical with Cohen's d value (0.15).

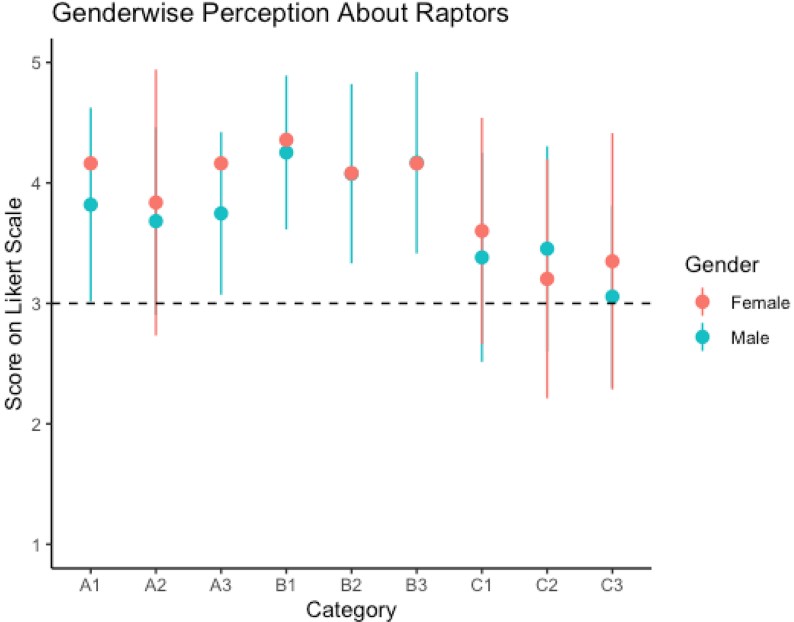

**Figure 2 Perception of male and female respondents towards raptors in arid region Rajasthan.** Error bar shows standard error. (A, Indirect benefits: A1, increases crop quality; A2, increases yield; A3, essential for crop production); (B, negative/disservice: B1, causes damage to pollinators; B2, causes damage to poultry; B3, causes damage to livestock); (C, direct benefits: C1, controls insects; C2, controls rodents; C3, alternative to pesticides); 1, very harmful; 2, harmful; 3, niether harmful nor beneficial; 4, beneficial; 5, very beneficial.

## Perception of disservices

There was no discernible difference between male and female respondents' perceptions of the detrimental effects of raptors on pollinators ($p = 0.173$), poultry ($p = 0.957$) and livestock ($p = 0.948$) (Fig. 2). Effect size was also merely statistical with Cohen's d value (0.13), (0.001) and (0.0001) respectively.

## Perception of direct benefits

There was a difference in the perception of males and females about the role of raptors in controlling rodents ($p = 0.013$) or insects ($p = 0.025$) and whether raptors could serve as an alternative to pesticides ($p = 0.002$) (Supporting Information S11). Cohen's d value for these categories were (0.27), (0.24) and (0.31) respectively which shows that difference in perception of male and female is merely statistical. Compared to conventional farmers, those growing organic crops had a positive perception towards raptors (Fig. 3). For both vultures and raptors, the regression plot (Fig. 4) shows that relation between the perception of respondents towards the species and the population trend of the species. The results of the factor analysis show that there are three broad factors can be used to divide the items on the Likert scale (Table 1). All the ecosystem services' benefits were adversely correlated with their disservices. For male respondents, the strength of the pairwise correlation was greater as compared to female respondents (Fig. 5). According to GLM analysis (Table 2), growing fruit crops and seed crops were the main factors influencing people's favourable attitudes toward raptors ($p = 0.02$ and $p = 0.001$ respectively). Chicken ownership and

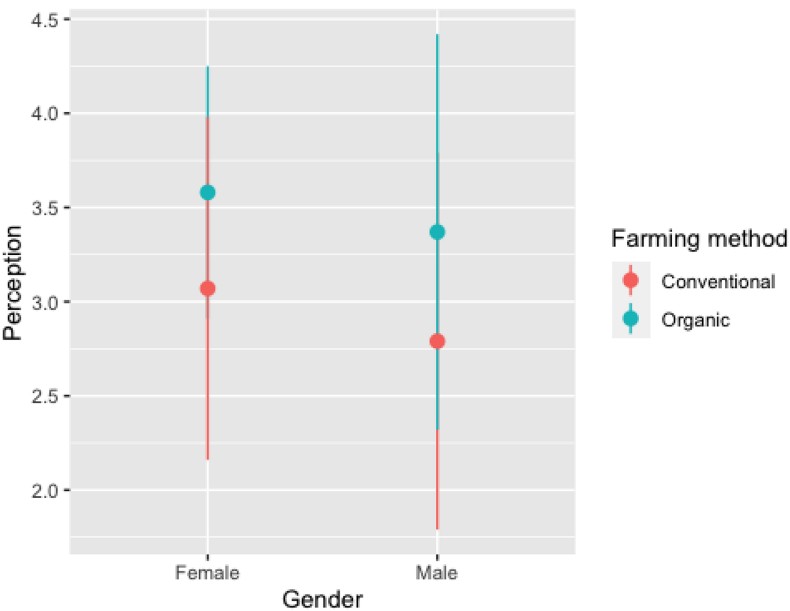

**Figure 3 Comparison of perception of raptors between respondents practicing conventional and organic farming, error bar shows standard error.** (1, Very harmful; 2, harmful; 3, niether harmful nor beneficial; 4, beneficial; 5, very beneficial).

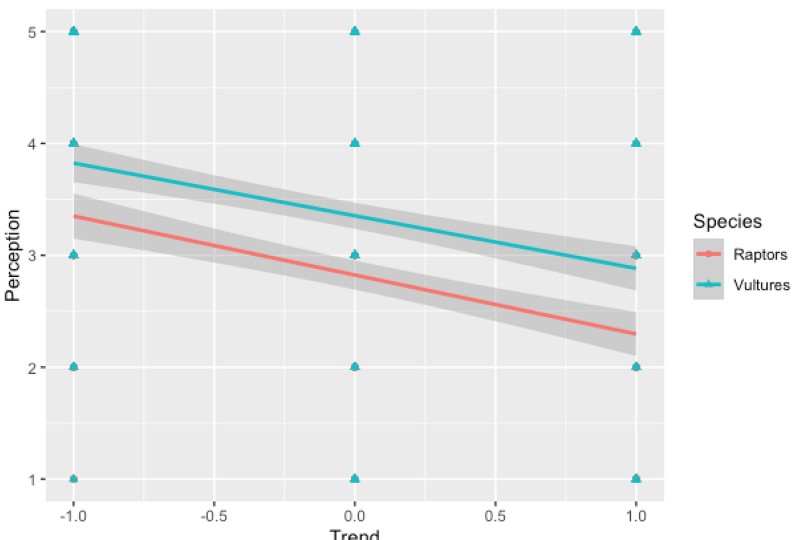

**Figure 4 Regression plot showing Perception of population of Raptors/Vultures *vs* Perception of Ecosystem services by Raptors/Vultures in arid region of Rajasthan.** X axis: −1, decreasing population; 0, stable population; 1, increasing population; Y axis: 1, very harmful; 2, harmful; 3, niether harmful nor beneficial; 4, beneficial; 5, very beneficial.

investing in chicken management were variables which were leading to a negative perception towards raptors (S12). Out of total 313 respondents who were growing fruits and seed crops in the region, 156 respondents expressed willingness in investing in methods which may help in raptor conservation like installing nest boxes, perches and providing suitable habitat for raptors in their field. Both male and female respondents felt

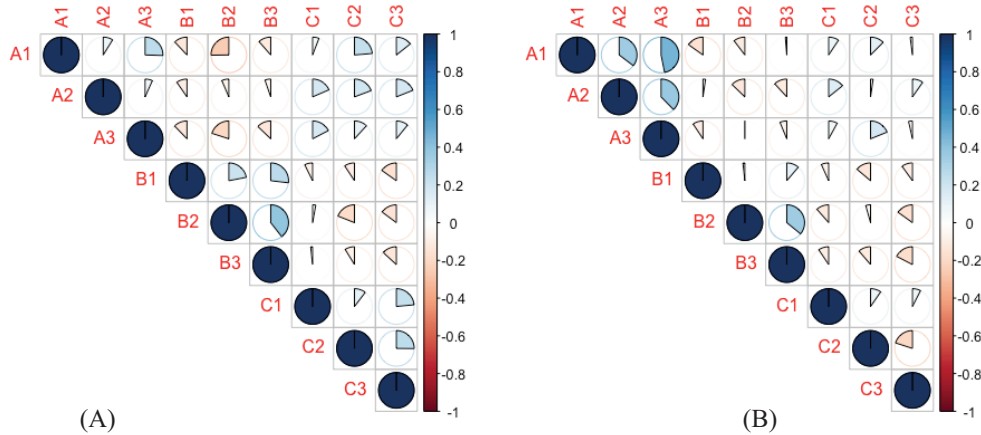

**Figure 5 Pairwise correlation between cultural ecosystem services as perceived by respondents ((A) Male, (B) Female) in arid region Rajasthan.** A, Indirect benefits, A1, increases crop quality; A2, increases yield; A3, essential for crop production. B, Disservices: B1, causes damage to pollinators; B2, causes damage to poultry; B3, causes damage to livestock. C, Direct benefits: C1, controls insects; C2, Controls Rodents; C3, alternative to pesticides.

that bats have positive impact on insect control but they dint affect crop yield much (S6). Perching birds were perceived to have a positive impact on tourism and insect control but were considered having negative impact on crop yield (S7).

## DISCUSSION

The concept of ecosystem services has gained widespread acceptance as a way for people to express the values they attach to different ecosystem functions (*Ferreira et al., 2018*). Perception studies can offer crucial information for observing, comprehending, and interpreting the social impacts and ecological results of conservation. If local perceptions towards a species or group of species which are found in one particular area is positive than it would help in conservation of that area (*Andrade & Rhodes, 2012*). Our findings, which demonstrate how farmers view raptors, highlights the need for ongoing research, focused outreach efforts, and legislative measures that give farmers the information which they need to choose for implementing wildlife-friendly agricultural practices (*Kross et al., 2016*) like (a) providing roosts, (b) managing habitat complexity, (c) reintroducing native species and (d) reducing impact of invasive species on target species (*Lindell et al., 2018*). Depending upon the impact they have, raptors are seen by the respondents as both beneficial and harmful. Raptors can help in pest control (*Raimilla & Rau, 2017*). On the other hand, they can cause negative perceptions due to poultry predation. It was also reported by some respondents who practiced poultry farming in the region. Apart from predation, raptors can also be a reason for transmission of diseases to poultry which may add to further losses (*Nyirenda et al., 2017*).

Additionally, raptors were viewed as being extremely beneficial for fruit growers' produce because they keep rodents and other pests off the farm. Putting up nest boxes to draw raptors can help reduce rodent populations on farms (*Coles, Wallis & Brennan, 2019*;

**Table 2 GLM analysis of Socio-Demographic variables and their effect on raptor perception in arid region Rajasthan.**

| Coefficient | Estimate | Std. error | Z value | Pr (>z) |
|---|---|---|---|---|
| (Intercept) | −16.99807 | 624.19538 | −0.027 | 0.978275 |
| Gender (Male) | 0.08132 | 0.36984 | 0.220 | 0.825974 |
| Fruit crops (growers) | −0.76781 | 0.33034 | −2.324 | 0.020111* |
| Farming.method (Conventional) | 1.08905 | 1.12765 | 0.966 | 0.334162 |
| Farming.method (Organic) | 0.20031 | 1.13471 | 0.177 | 0.859876 |
| Education (Higher secondary) | 14.83957 | 624.19389 | 0.024 | 0.981033 |
| Education (Secondary) | 13.51015 | 624.19390 | 0.022 | 0.9827 |
| Education (Primary) | 14.13420 | 624.19389 | 0.023 | 0.9819 |
| Education (None) | 13.53437 | 624.19417 | 0.022 | 0.9827 |
| Seed crops (Growers) | −1.67898 | 0.43777 | −3.835 | 0.000125* |
| Forage crops (Growers) | 1.55572 | 0.40745 | 3.818 | 0.0546 |
| Vegetable (Growers) | 0.55477 | 0.50588 | 1.097 | 0.272792 |
| Livestock (Owners) | 0.83013 | 0.76202 | 1.089 | 0.275983 |
| Null deviance: 501.08 on 372 degrees of freedom | | | | |
| Residual deviance: 422.16 on 360 degrees of freedom | | | | |
| AIC: 448.16, Number of fisher scoring iterations: 13 | | | | |

**Note:**
 An asterisk (*) indicates a significant value.

*Paz Luna et al., 2020*). Frugivory by raptors is reported by *Fitzsimons & Leighton (2021)* but in our study region, no such case was confirmed by the respondents.

 We observed that both male and female organic farmers had positive perception of raptors and were willing to spend for their conservation. Among respondents, most of the organic farmers believed that their cropping method could also help in the conservation of raptors, it was also observed by *Kirk, Martin & Freemark Lindsay (2020)*. Integrated pest management (IPM) is a decision-based process involving coordinated use of multiple tactics for optimizing the control of all classes of pests (insects, pathogens, weeds, vertebrates) in an ecologically and economically sound manner. It involves regular monitoring of pests, and their natural enemies (*Ehler, 2006*). Raptors play an effective role in controlling damage to crops by feeding on pests (*Peisley, Saunders & Luck, 2017*; *Gorosábel, Bernad & Pedrana, 2022*) and are important part of IPM (*Zagorski, 2019*). The rodent population can be controlled by conserving raptors (*Antkowiak & Hayes, 2004*), constructing nest boxes (*Paz Luna et al., 2020*) or by erecting artificial perches (*Kay et al., 1994*), which in long term will also lower the cost of farming inputs on pesticides and rodenticides (*Machar et al., 2017*). They can serve as effective alternatives to pesticides and reduce the impact which these harmful chemicals have on food chain (*María et al., 1996*; *Hughes et al., 2013*). As conventional farming is more common than organic farming, further research in this region is needed to understand role of the raptors for controlling insect pests.

 Views of female respondents on the effect of raptors on livestock varied quite significantly from those of males as most of the female respondents spent more time with

their livestock/poultry in the sheds. They spent less time outdoors as compared to male counterparts.

According to research conducted by *Courchamp et al. (2006)* and *Hall, Milner-Gulland & Courchamp (2008)*, the perception of raptors and vultures as beneficial species is influenced by the level of rarity attributed to these species in terms of their perceived population. They found a positive relationship between species rarity and perception, indicating that species perceived as rarer are often viewed more positively.

In contrast, *Morales-Reyes et al. (2018)* discovered an opposite relationship between species rarity and the perception of species as providers of ecosystem services. This suggests that as species become rarer, they may be perceived as less capable of providing ecosystem services. These studies shed light on the complex relationship between species rarity, perception, and the evaluation of their role in ecological processes. The general public gives more value to rare species relative to common ones (*Angulo & Courchamp, 2009*). It is a common belief that attitudes and perceptions towards a species are influenced by the degree of its rarity. Although, it was reported that rareness in terms of distribution cannot be a criterion in the decision for investing on conservation of the species (*Martín-López, Montes & Benayas, 2007*). Elusive species which are globally considered as endangered and are least known are rarely perceived as emblematic (*Cortés-Avizanda et al., 2018*) but our results on rarity and perception towards a species are in concurrence with study done by *Otsuka, Nakano & Takahashi (2016)* which indicates that farmers have a species-specific view that incorporates cultural and aesthetic value of rare species, and they prefer usefulness of these species over other.

The variables were interpreted as various categories that stood in for various ecosystem benefits and drawbacks. Disservices were found to be negatively correlated with ecosystem services and other categories for both male and female respondents' negative correlations between disservices and other cultural ecosystem services suggest that these categories are dependent on each other. People are influenced by general positive or negative effects when judging disservices and benefits. It suggests that likeability of respondents towards raptors was positively correlated with direct and indirect benefits while negatively correlated with the disservices. This "Dr. Jekyll and Mr. Hyde" paradox (*Morales-Reyes et al., 2018*) can be understood by socio economic characteristics of the respondents who are involved in poultry management. They perceive raptor predation of chickens as a loss to their livelihood. Also, livestock owners view raptors as a threat to the newborn cattle and a carrier of disease. On the other hand, fruit growing and seed growing farmers and those practicing organic agriculture perceive raptors as beneficial in their effect of controlling rodents and pests. Strength of correlation was slightly higher for the male respondents. It may be explained by the fact that in this region male respondents are more involved in farming, working as nature guides and transhumance and they interact with raptors more often than female respondents.

Implementing long term conservation plans needs taking social perspective in consideration, misled perception of a species can be detrimental for its survival (*Ceríaco, 2012*). To change farmers' behaviours toward more sustainable conservation of farmland biodiversity, instruments should aim to influence individual farmer's motivation and

behaviour. However, a lack of knowledge of farmers' opinions toward wildlife can lead to poor integration of conservation measures (*Katuwal et al., 2021*; *Kross et al., 2018*). We should aim to place farmland biodiversity "in the hands and minds" of farmers (*Ahnström, 2009*). Without an appreciation of the human dimension to problems of conflict, sustaining species outside protected areas may be difficult (*Lee & Priston, 2005*).

## CONCLUSION

Conservation of raptors requires landscape-based approach beyond the protected areas. Very few resources and funding are allocated for the conservation of the raptors residing outside protected areas. Arid region of Rajasthan is home of many species of raptors but the overall conservation planning for raptors needs to include a socio-ecological perspective. Designing education and awareness programs along with community participation can reduce conflict with raptor in rural regions and will be beneficial for implementation of long-term conservation programs.

## ACKNOWLEDGEMENTS

We are grateful to the Director, Amity Institute of Forestry and Wildlife, Amity University, Noida, for their encouragement. We are also grateful to Mr. Vinay Solanki (Judicial Magistrate, Sujangarh) for introducing us to the landscape, helping us in logistics and stay during field work and making our survey work smooth.

### Funding

The authors received no funding for this work.

### Competing Interests

Randeep Singh is an Academic Editor for PeerJ. Rahul Kaul is employed by the Wildlife Trust of India. Puneet Pandey is employed by the Conservation Genome Resource Bank for Korean Wildlife (CGRB), Research Institute for Veterinary Science and College of Veterinary Medicine, Seoul National University, Seoul, Republic of Korea, and is an affiliate of Enprotec India Foundation.

### Author Contributions

- Govind Tiwari conceived and designed the experiments, performed the experiments, analyzed the data, prepared figures and/or tables, data collection and field work, and approved the final draft.
- Puneet Pandey analyzed the data, prepared figures and/or tables, authored or reviewed drafts of the article, and approved the final draft.
- Rahul Kaul conceived and designed the experiments, analyzed the data, authored or reviewed drafts of the article, and approved the final draft.
- Randeep Singh conceived and designed the experiments, analyzed the data, prepared figures and/or tables, authored or reviewed drafts of the article, and approved the final draft.

## Human Ethics

The following information was supplied relating to ethical approvals (*i.e.*, approving body and any reference numbers):

This study was approved by the Research Ethics Committee of Amity University Uttar Pradesh (AUUP/AIFWREC/2020/01).

## Data Availability

The code and raw data are available in the Supplemental Files.

## Supplemental Information

Supplemental information for this article can be found online at http://dx.doi.org/10.7717/peerj.15996#supplemental-information.

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
