# Peer review of "Farmers’ perception of the ecosystem services provided by diurnal raptors in arid Rajasthan"

_PeerJ, doi:10.7717/peerj.15996_

## Round 0.1 · original submission · Major Revisions

The two reviewers I have consulted gave a somewhat different assessment of the manuscript, with the first reviewer being more positive and the second one raising more fundamental issues. Having carefully evaluated the submission myself, I decided to go along with the recommendation of reviewer #1 and request major revisions in light of the potential value of the paper. Please take into consideration the fact that both reviewers pointed out the need for a more detailed elaboration on the analyses you performed and methods used. In case you decide to submit a revised version, I am likely to seek the opinions of both reviewers again. Finally, research on human subjects (including interviewing people) generally requires approval from a human ethics committee: please make sure to mention whether such approval was obtained or, if not needed in your institution, to explicitly make explicit mention of this.

·

Basic reporting

Overall, I think this is a really interesting paper providing valuable information about farmer perceptions in a region where this hasn't previously been done. I think that more information on the raptors found in the study region would be helpful, especially citing any papers that have recorded raptor consumption of crop pests or of livestock/poultry. The paper could benefit from a bit more proofreading- but is generally easy to follow and well-written.

I have suggested some additional papers on the role of raptors in farming systems in the marked up PDF attached, which you may find helpful to add to the paper.

There are some areas where text that belongs in the results is in the discussion and vice-versa.
I was the lead author on the Kross et al. 2018 paper you cite on farmer perceptions of wildlife- and this paper looks very similar in its approach, hypotheses, taxa of interest, and the questions asked (although ours was based on written surveys and yours on interviews). It's so exciting to see a study that's so similar being done in a different region! However, if you did base your study on ours, it would be great to see that explicitly stated in the methods section- and it also gives you an interesting point of comparison between farmers in the two regions for the discussion. Your hypotheses might have also been based on the findings from that paper (which is cited already).

Experimental design

I think a bit of information in the methods on how you recruited and selected farmers for the study is needed. Were all farmers independent of one another in some way? Did you ever interview a male and female farmer from the same farm? Why was there a gap in the surveying period (or did you specifically separate it in two periods?).
I would like to know how you designed the models that were used, which response variables were used in each model, how you checked the models, and any steps you took to simplify the models. The model output tables also need a bit more detail (see the marked up PDF for some suggestions for additional information to include).

Validity of the findings

The findings are a little hard to interpret in this version of the paper as the key effect sizes aren't reported and can't be easily deciphered from the figures or tables. It would be really helpful to report mean and standard errors for all of the results you mention in the text, and to include such information in the abstract and discussion as well.
I made some suggestions on the attached PDF for the figures- offsetting the male/female bars is needed to see the spread of the data- and it's not clear from these tables how different these groups were.
Figure 7 is not mentioned in the text- can you point to this in the results and interpret it?
Supporting information figures 3 and 4 have the same legends- are they the same plots twice?
S5, S6, S7 all need to include some measure of variance- I suggest standard error, but you could use another.

Additional comments

I think your finding on the perceptions of changes in abundance and the perceptions of ecosystem services is really interesting- it's worth adding that to the abstract as a major finding. However, did you ask this question for every species of raptor in the region and then ask for an ecosystem service score? If yes- that level of detail would be great to include in the paper and results- and it would be interesting to match those factors to things like diet and body size (like the Echeverri paper does) or at least to discuss that. Is an alternative explanation for the finding that the species that actually do provide more ecosystem services are in decline?

·

Basic reporting

I have made several annotations in the .pdf of the manuscript. In my view though parts of the article are clearly worded such as the methods and data analysis, the authors fail to use a clear and unambiguous English in the theoretical parts such as the Introduction and Discussion sections. Although the literature references are sufficient, in some places as indicated in the .pdf a bit more elaboration is required. Simple mistakes in the figures that I could see have been indicated. I would also like to state that I felt that the authors did not note the statements correctly. This has resulted in the authors putting up contradictory statements in the article (these have also been indicated in the .pdf).

Experimental design

I could not understand if the supplement file 1 was the only questions that were asked and if the interviewee's responses, if they were expansive, which might have seemed outside the purview of the questionnaire were recorded by the researchers in some other form. The authors do not seem to realize the ambiguity or relativity that opinion based "social sciences" research can have. From my reading I got a sense that the authors have used mixed methods research without properly assessing or gaining understanding of these methods. Therefore though they have used the statistical methods rigorously certain statements which probably were not part of the questionnaire have been reported here and there without proper indications of their origins.

Validity of the findings

No comment.

---

## Round 0.2 · Major Revisions

Thank you for the work you have done in revising and resubmitting the manuscript. The two reviewers I have consulted both appreciated the revisions and clarifications you have made and felt that the manuscript has substantially improved compared to the original version. Nevertheless, both reviewers and in particular Reviewer #1, still raise a number of questions and comments which are fairly major and will require an additional revision before the manuscript can be further considered for publication.

·

Basic reporting

Kudos to the authors for dealing with the many revisions I and the other reviewer asked for. I think that overall this manuscript is much clearer. There are a few minor changes I would suggest making to the language, but I think that the points brought up below may need to be addressed first (also I was adding comments/suggestions to the tracked changes version of the manuscript, but then realized it was different from the pdf of the revised manuscript).

Experimental design

However, in reading through the latest version, I’m still unclear on which analyses you used- and I’m mainly confused about what you mean by ‘perception’ as it pertains to your survey instrument and data analyses. I think it would be very helpful for the reader to know what data was used for each analysis. You collected data from a likert-scale, and the figures are presented on a similar scale, but maybe I’m misinterpreting exactly what data was used for the figures. When you talk about perceptions, are you talking about the likert-scale responses, or to the factor analyses?
• Table 2- Why wasn’t the variable for poultry farming included in the GLM for the variables on raptor perception? Or is ‘livestock’ the same as poultry farming? Please add to the table legend that it was a binomial GLM and what the response variable was. Mention also what the intercept represents (it looks like female farmers, but what was the baseline for the other categorical variables- what was the baseline for crop type? Farming method? Education?). I would also mention that these are untransformed model output.
• Looking at the questionnaire- you were asking people to agree or disagree with the statements about whether raptors were beneficial to them- along with a list that already ‘leads’ them to consider benefits or costs. They were then asked to agree or disagree with the statement- so it appears that the lower Likert values for the ‘disservice’ sections (B1-B3) were actually that respondents were disagreeing with the statement that raptors cause damage to pollinators/poultry/livestock.
• Figure 2- I don’t really understand this figure and am not sure what it adds for the reader. Perhaps if you added in the specific analytical steps you took and how the data was converted (e.g. for a score of ‘perception’) that would be helpful.
• Figure 4- explain briefly in the figure legend how the perception score was calculated for this comparison. Again, is this showing standard error or some other measure of variance?
• Figure 6- Needs more information in the figure legend. Why are there 2 panels? These aren’t all cultural ecosystem services right?
• Line 174/Supplementary figures S3-S4- what was this used for? What were the factors that went into the analysis for this? How many factors did you then use?
• Table 1- what are the 3 columns for ‘factor loading’ representative of? Explain in the table legend or at the top of the columns.
• Table S1- it would be good if the data in columns 2:5 was actually presented for males and females separately.

Validity of the findings

• I do still think it’s important that you highlight your key effect sizes in the text of the results, rather than just reporting p-values. Counting on a reader to find and open the last table in the supplementary material to find the estimates for your ANOVA is asking too much. All of the p-values presented on lines 203-210 would be considered significant.
• I became a little confused with the results that were presented specifically for vultures in Figure 5, but aren’t presented separately for the other analyses. Were the respondents asked the survey questions regarding raptors, then vultures, then bats, then birds? The farmer perception questionnaire currently includes no mention of vultures, so it isn’t clear how that data was derived. If respondents were asked about raptors separately of vultures, was it made clear to them what you meant when you asked about raptors that it did not include vultures? What about owls- were they instructed to think of just diurnal raptors?
• I’d recommend you convert all of your standard deviations to standard errors. Looking at Figure 3, it appears from the error bars that you shouldn’t expect any significant differences between male and female perceptions of raptors- but I think you’re using the standard deviations on that figure. Standard error should bring those error bars back and reflect the significant results you found for some of the comparisons.
• The text currently says nothing about the direction of the responses (whether they were more positive or more negative). Was 3 ‘Neutral’ on your Likert scale (line 154)? I recommend adding a neutral line across figure 3 to show readers where participants were more positive or more negative towards raptors in each category. I’d also recommend you label the y-axis to show what the numbers represent on the Likert scale, and allowing the scale to go to 5, since that was the top value.
• Line 210- why not include the results for organic vs. conventional perceptions for the Indirect Benefits and Detrimental services sections? (also, line 203- stay consistent with labels, this section is called ‘Negative/Disservice’ in Figure 3’s legend)
• Line 269- Can this be rephrased. It isn’t clear here if the general perception is that as species become more rare they are perceived as more important for ecosystem services, or that as a species becomes more rare they are perceived as providing fewer services. This paragraph generally is a bit hard to follow.

·

Basic reporting

I feel the language can be further improved, however I also agree that sufficient literature, field and background context has been provided in the article.

Experimental design

the experimental design seems amicable and address a pertinent conservation policy issue.

Validity of the findings

the findings would be helpful in designing intervention measures for raptor conservation in agricultural landscapes.

Additional comments

no comment

---

## Round 0.3 · Minor Revisions

I am glad to report that the reviewer who has examined the resubmitted version of your manuscript finds that it should become acceptable for publication after taking care of a few minor remaining issues. Please see their assessment below, which I agree with. In addition, please ensure that the manuscript is fully proofread before the final submission. I highlight here below a few editing issues that remain to be addressed (but there may be others I overlooked).

- L43: “Social” should not be capitalized
- L46: replace “decision” with “decisions”. Also, remove the full stop after “decisions”.
- L47: “contributes to” with “constitutes”
- L52: “encourages” with “encourage”. Check the spacing at the beginning of this sentence and before the Kross et al reference.
- L81: add comma after “extent”
- L85-87: This sentence need rephrasing. Also, at L87, replace “peoples’” with “people’s”
- L91: “believed” with “believe”
- L92: “are living” with “live”
- L82-92: It is fine that you chose to focus on sense of place and education, but please include a mention somewhere in this paragraph that there are other cultural ecosystem services beyond those two.
- L123: “was” with “is”
- L134: “allied” with “related”
- L143: “in person” with “in-person”
- L152: “agree 5-Strongly” with “5-Strongly agree”
- L168: “likert” with “Likert”. Same at L169 and L174.
- L182: “Cohen’ s” with “Cohen’s”
- L184-192: the terms “explanatory” and “predictive” variables are used in a somewhat confusing way in this paragraph. I suggest replacing with “dependent” variable (to indicate the binary variable you are trying to predict) and “independent” variables for the variables you use as explanatory variables of the model.
- L199: please be explicit (either here or, better, in the methodology section) as to what p-value threshold you consider to determine statistical significance. I assume p=0.05 but I do not see that this is explicitly stated anywhere.
- L219: the wording in “ecosystem services advantages” and “disadvantages” is somewhat awkward. Perhaps replace with “benefits” and “ecosystem disservices”?
- L228: “dint” with “do not”
- L270: “that” with “those”
- L271: “spent” with “spend”
- L269-284: while I get the general conclusion at the end of this paragraph, I find it hard to follow the reasoning here (e.g., when you discuss studies finding “positive relationship” and “negative relationship”). I suggest rephrasing for clarity.
- L285: which “variables”?
- L287: I believe there should be a full stop after “respondents” with the new sentence starting with “Negative correlations”.
- L292: “socio economic” with “socio-economic”
- L293: “told” does not seem like the correct word here. Maybe “perceive”?
- L297: “more” with “higher”.
- L299-300: replace “their interaction with raptors is more as compared to female respondents” with “they interact with raptors more often than female respondents”
- L302: not sure in what sense a perception can be right or “wrong”. Maybe replace with “misguided” or “misled”.

·

Basic reporting

I think that the paper is significantly clearer and am happy with the changes made by the authors. I recommend that the editorial team read through for some basic editing re: typos, etc.

Experimental design

The changes are clear and I'm glad that the scales were corrected for some of the disservices points.

Validity of the findings

I think the only points that need clarification are that the type of error bars are named in the figure legends, and that anywhere a p-value is cited, the effect size or mean for that value is also provided.

---

## Round 0.4 · Minor Revisions

Thank you for addressing my comments. Please note that Reviewer #1 also had two remaining comments on your previous submission, which do not appear to have been addressed. I am pasting them below, for your convenience. Please address these as well so that I can proceed further with the manuscript.

Comments from Reviewer #1:
"I think the only points that need clarification are that the type of error bars are named in the figure legends, and that anywhere a p-value is cited, the effect size or mean for that value is also provided."

---

## Round 0.5 · Minor Revisions

Thank you for addressing the last remaining comments. Congratulations on this interesting work!

Before this can be accepted, there is an administrative matter that needs to be attended to. Please see the staff note below:

---

## Round 0.6 · accepted · Accept

Thank you for providing the approval from the human ethics committee and again congratulations for this interesting work.